

# Adaptive divergence for rapid adversarial optimization

Maxim Borisyak[1], Tatiana Gaintseva[1] and Andrey Ustyuzhanin[1,2]

[1] Laboratory of Methods for Big Data Analysis, National Research University Higher School of Economics, Moscow, Russia
[2] Physics Department, Imperial College, London, United Kingdom

Corresponding author
Maxim Borisyak, mborisyak@hse.ru

## ABSTRACT

Adversarial Optimization provides a reliable, practical way to match two implicitly defined distributions, one of which is typically represented by a sample of real data, and the other is represented by a parameterized generator. Matching of the distributions is achieved by minimizing a divergence between these distribution, and estimation of the divergence involves a secondary optimization task, which, typically, requires training a model to discriminate between these distributions. The choice of the model has its trade-off: high-capacity models provide good estimations of the divergence, but, generally, require large sample sizes to be properly trained. In contrast, low-capacity models tend to require fewer samples for training; however, they might provide biased estimations. Computational costs of Adversarial Optimization becomes significant when sampling from the generator is expensive. One of the practical examples of such settings is fine-tuning parameters of complex computer simulations. In this work, we introduce a novel family of divergences that enables faster optimization convergence measured by the number of samples drawn from the generator. The variation of the underlying discriminator model capacity during optimization leads to a significant speed-up. The proposed divergence family suggests using low-capacity models to compare distant distributions (typically, at early optimization steps), and the capacity gradually grows as the distributions become closer to each other. Thus, it allows for a significant acceleration of the initial stages of optimization. This acceleration was demonstrated on two fine-tuning problems involving Pythia event generator and two of the most popular black-box optimization algorithms: Bayesian Optimization and Variational Optimization. Experiments show that, given the same budget, adaptive divergences yield results up to an order of magnitude closer to the optimum than Jensen-Shannon divergence. While we consider physics-related simulations, adaptive divergences can be applied to any stochastic simulation.

## INTRODUCTION

Adversarial Optimization (AO), introduced in Generative Adversarial Networks (*Good-fellow et al., 2014*), became popular in many areas of machine learning and beyond with applications ranging from generative (*Radford, Metz & Chintala, 2015*) and inference

tasks (*Dumoulin et al., 2016*), improving image quality (*Isola et al., 2017*) to tuning stochastic computer simulations (*Louppe, Hermans & Cranmer, 2017*).

AO provides a reliable, practical way to match two implicitly defined distributions, one of which is typically represented by a sample of real data, and the other is represented by a parameterized generator. Matching of the distributions is achieved by minimizing a divergence between these distribution, and estimation of the divergence involves a secondary optimization task, which, typically, requires training a model to discriminate between these distributions. The model is referred to as discriminator or critic (for simplicity, we use term discriminator everywhere below).

Training a high-capacity model, however, is computationally expensive (*Metz et al., 2016*) as each step of divergence minimization is accompanied by fitting the discriminator; therefore, adversarial training often requires significantly more computational resources than, for example, a classification model with a comparable architecture of the networks.[1] Nevertheless, in conventional settings like GAN, this problem is not pronounced for at least two reasons. Firstly, the generator is usually represented by a deep neural network, and sampling is computationally cheap; thus, for properly training the discriminator, a sample of a sufficient size can be quickly drawn. Secondly, GAN training procedures are often regarded not as minimization of a divergence, but as game-like dynamics (*Li et al., 2017*; *Mescheder, Geiger & Nowozin, 2018*); such dynamics typically employ gradient optimization with small incremental steps, which involve relatively small sample sizes for adapting the previous discriminator to an updated generator configuration.

Computational costs of AO become significant when sampling from the generator is computationally expensive, or optimization procedure does not operate by performing small incremental steps (*Metz et al., 2016*). One of the practical examples of such settings is fine-tuning parameters of complex computer simulations. Such simulators are usually based on physics laws expressed in computational mathematical forms like differential or stochastic equations. Those equations relate input or initial conditions to the observable quantities under conditions of parameters that define physics laws, geometry, or other valuable property of the simulation; these parameters do not depend on inputs or initial conditions. It is not uncommon that such simulations have very high computational complexity. For example, the simulation of a single proton collision event in the CERN ATLAS detector takes several minutes on a single core CPU (*The ATLAS Collaboration, 2010*). Due to typically high dimensionality, it takes a considerable amount of samples for fine-tuning, which in turn increases the computational burden.

Another essential property of such computer simulations is the lack of gradient information over the simulation parameters. Computations are represented by sophisticated computer programs, which are challenging to differentiate.[2] Thus, global black-box optimization methods are often employed; Bayesian Optimization is one of the most popular approaches.

In this work, we introduce a novel family of divergences that enables faster optimization convergence measured by the number of samples drawn from the generator. The variation of the underlying discriminator model capacity during optimization leads to a significant speed-up. The proposed divergence family suggests using low-capacity models to compare

[1] For instance, compare training times, network capacities and computational resources reported by *Simonyan & Zisserman (2014)* and *Choi et al. (2018)*.

[2] There are ways to estimate gradients of such programs, for example, see (*Baydin et al., 2019*). However, all methods known to the authors require training a surrogate, which encounters the problem of the expensive sampling procedures mentioned above.

distant distributions (typically, at early optimization steps), and the capacity gradually grows as the distributions become closer to each other. Thus, it allows for a significant acceleration of the initial stages of optimization. Additionally, the proposed family of divergences is broad, which offers a wide range of opportunities for further research.

We demonstrate the basic idea with some toy examples, and with a realistic challenge of tuning Pythia event generator (*Sjöstrand, Mrenna & Skands, 2006*; *Sjostrand et al., 2015*) following *Louppe, Hermans & Cranmer (2017)* and *Ilten, Williams & Yang (2017)*. We consider physics-related simulations; nevertheless, all proposed methods are simulation-agnostic.

## BACKGROUND

Adversarial Optimization, initially introduced for Generative Adversarial Networks (GAN) (*Goodfellow et al., 2014*), offers a general strategy for matching two distributions. Consider feature space $\mathcal{X}$, ground-truth distribution $P$, and parametrized family of distributions $Q_\psi$ implicitly defined by a generator with parameters $\psi$. Formally, we wish to find such $\psi^*$, that $P = Q_{\psi^*}$ almost everywhere. AO achieves that by minimizing a divergence or a distance between $P$ and $Q_\psi$ with respect to $\psi$. One of the most popular divergences is Jensen–Shannon divergence:

$$
\begin{aligned}
\mathrm{JSD}(P, Q_\psi) &= \frac{1}{2}\Big[\mathrm{KL}(P\|M_\psi) + \mathrm{KL}(Q_\psi\|M_\psi)\Big] \\
&= \frac{1}{2}\mathop{\mathbb{E}}_{x\sim P}\log\frac{P(x)}{M_\psi(x)} + \frac{1}{2}\mathop{\mathbb{E}}_{x\sim Q_\psi}\log\frac{Q(x)}{M_\psi(x)};
\end{aligned}
\tag{1}
$$

where: KL —Kullback–Leibler divergence, $M_\psi(x) = \frac{1}{2}\big(P(x) + Q_\psi(x)\big)$. The main insight of *Goodfellow et al. (2014)* is that JSD can be estimated by training a discriminator $f$ to distinguish between $P$ and $Q_\psi$:

$$
\begin{aligned}
&\log 2 - \min_{f\in\mathcal{F}} L(f, P, Q_\psi) = \\
&\quad \log 2 - \min_{f\in\mathcal{F}}\left\{-\frac{1}{2}\mathop{\mathbb{E}}_{x\sim P}\log\big(f(x)\big) - \frac{1}{2}\mathop{\mathbb{E}}_{x\sim Q_\psi}\log\big(1-f(x)\big)\right\} = \\
&\quad \log 2 + \left\{\frac{1}{2}\mathop{\mathbb{E}}_{x\sim P}\log\big(f^*(x)\big) + \frac{1}{2}\mathop{\mathbb{E}}_{x\sim Q_\psi}\log\big(1-f^*(x)\big)\right\} = \\
&\quad \log 2 + \left\{\frac{1}{2}\mathop{\mathbb{E}}_{x\sim P}\log\frac{P(x)}{Q_\psi(x)+P(x)} + \frac{1}{2}\mathop{\mathbb{E}}_{x\sim Q_\psi}\log\frac{Q_\psi(x)}{Q_\psi(x)+P(x)}\right\} = \\
&\quad\quad \frac{1}{2}\mathop{\mathbb{E}}_{x\sim P}\log\frac{P(x)}{M_\psi(x)} + \frac{1}{2}\mathop{\mathbb{E}}_{x\sim Q_\psi}\log\frac{Q(x)}{M_\psi(x)} = \mathrm{JSD}(P, Q_\psi); \quad (2)
\end{aligned}
$$

where: $L$ —cross-entropy loss function, $\mathcal{F} = \{f : \mathcal{X} \to [0, 1]\}$ is the set of all possible discriminators, and $f^*$ is the optimal discriminator. Similar formulations also exist for other divergences such as Wasserstein (*Arjovsky, Chintala & Bottou, 2017*) and Cramer (*Bellemare et al., 2017*) distances.

In classical GAN, both generator and discriminator are represented by differentiable neural networks. Hence, a subgradient of $\mathrm{JSD}(P, Q_\psi)$ can be easily computed (*Goodfellow*

*et al., 2014*). The minimization of the divergence can be performed by a gradient method, and the optimization procedure goes iteratively following those steps:

- using parameters of the discriminator from the previous iteration as an initial guess, adjust $f$ by performing several steps of the gradient descent to minimize $\mathcal{L}(f, P, Q_\psi)$;
- considering $f$ as a constant, compute the gradient of $\mathcal{L}(f, P, Q_\psi)$ w.r.t. $\psi$, perform one step of the gradient ascent.

For computationally heavy generators, gradients are usually practically unfeasible; therefore, we consider black-box optimization methods. One of the most promising methods for black-box AO is Adversarial Variational Optimization (*Louppe, Hermans & Cranmer, 2017*), which combines AO with Variational Optimization (*Wierstra et al., 2014*). This method improves upon conventional Variational Optimization (VO) over Jensen–Shannon divergence by training a single discriminator to distinguish samples from ground-truth distribution and samples from a mixture of generators, where the mixture is defined by the search distribution of VO. This eliminates the need to train a classifier for each individual set of parameters drawn from the search distribution.

Bayesian Optimization (BO) (*Mockus, 2012*) is another commonly used black-box optimization method, with applications including tuning of complex simulations (*Ilten, Williams & Yang, 2017*). As we demonstrate in 'Experiments, BO can be successfully applied for Adversarial Optimization.

## ADAPTIVE DIVERGENCE

Notice, that in equation Eq. (2) minimization is carried over the set of all possible discriminators $\mathcal{F} = \{f : \mathcal{X} \mapsto [0, 1]\}$. In practice, this is intractable and set $\mathcal{F}$ is approximated by a model such as Deep Neural Networks. Everywhere below, we use terms 'low-capacity' and 'high-capacity' to describe the set of feasible discriminator functions: low-capacity models are either represent a narrow set of functions (e.g., logistic regression, shallow decision trees) or are heavily regularized (see 'Implementation' for more examples of capacity regulation); high-capacity models are sufficient for estimating JSD for an Adversarial Optimization problem under consideration.

In conventional GAN settings, the generator is represented by a neural network, sampling is computationally cheap, and usage of high-capacity discriminators is satisfactory. In our case, as was discussed above, simulations tend to be computationally heavy, which, combined with a typically slow convergence of black-box optimization algorithms, might make AO with a high-capacity model practically intractable.

The choice of the model has its trade-off: high-capacity models provide good estimations of JSD, but, generally, require large sample sizes to be properly trained. In contrast, low-capacity models tend to require fewer samples for training; however, they might provide biased estimations. For example, if the classifier is represented by a narrow set of functions $M \subseteq \mathcal{F}$, then quantity:

$$D_M(P, Q) = \log 2 - \min_{f \in M} L(f, P, Q); \qquad (3)$$

might no longer be a divergence, so we refer to it as *pseudo-divergence.*

**Definition 1** *A function* $D : \Pi(\mathcal{X}) \times \Pi(\mathcal{X}) \to \mathbb{R}$ *is a pseudo-divergence, if*:

**(P1)** $\forall P, Q \in \Pi(\mathcal{X}) : D(P, Q) \geq 0$;

**(P2)** $\forall P, Q \in \Pi(\mathcal{X}) : (P = Q) \Rightarrow D(P, Q) = 0$; *where* $\Pi(\mathcal{X})$ —*set of all probability distributions on space* $\mathcal{X}$.

It is tempting to use a pseudo-divergence $D_M$ produced by a low-capacity model $M$ for Adversarial Optimization, however, a pseudo-divergence might not guarantee proper convergence as there might exist such $\psi \in \Psi$, that $\text{JSD}(P, Q_\psi) > 0$, while $D(P, Q_\psi) = 0$. For example, naive Bayes classifier is unable to distinguish between $P$ and $Q$ that have the same marginal distributions. Nevertheless, if model $M$ is capable of distinguishing between $P$ and some $Q_\psi$, $D_M$ still provides information about the position of the optimal parameters in the configuration space $\psi^*$ by narrowing search volume, *Ilten, Williams & Yang (2017)* offers a good demonstration of this statement.

The core idea of this work is to replace Jensen–Shannon divergence with a so-called adaptive divergence that gradually adjusts model capacity depending on the 'difficulty' of the classification problem with the most 'difficult' problem being distinguishing between two equal distributions. Formally, this gradual increase in model complexity can be captured by the following definitions.

**Definition 2** *A family of pseudo-divergences* $\mathcal{D} = \{D_\alpha : \Pi(\mathcal{X}) \times \Pi(\mathcal{X}) \to \mathbb{R} | \alpha \in [0, 1]\}$ *is ordered and complete with respect to Jensen–Shannon divergence if*:

**(D0)** $D_\alpha$ *is a pseudo-divergence for all* $\alpha \in [0, 1]$;

**(D1)** $\forall P, Q \in \Pi(\mathcal{X}) : \forall 0 \leq \alpha_1 < \alpha_2 \leq 1 : D_{\alpha_1}(P, Q) \leq D_{\alpha_2}(P, Q)$;

**(D2)** $\forall P, Q \in \Pi(\mathcal{X}) : D_1(P, Q) = \text{JSD}(P, Q)$.

There are numerous ways to construct a complete and ordered w.r.t. JSD family of pseudo-divergences. In the context of Adversarial Optimization, we consider the following three methods. The simplest one is to define a nested family of models $\mathcal{M} = \{M_\alpha \subseteq \mathcal{F} | \alpha \in [0, 1]\}$, (e.g., by changing number of hidden units of a neural network), then use pseudo-divergence Eq. (3) to form a desired family.

Alternatively, for a parameterized model $M = \{f(\theta, \cdot) | \theta \in \Theta\}$, one can use a regularization $R(\theta)$ to control 'capacity' of the model:

$$D_\alpha(P, Q) = \log 2 - L(f(\theta^*, \cdot), P, Q); \tag{4}$$

$$\theta^* = \arg\min_{\theta \in \Theta} L(f(\theta, \cdot), P, Q) + c(1 - \alpha) \cdot R(\theta);$$

where $c : [0, 1] \to [0, +\infty)$ is a strictly increasing function and $c(0) = 0$.

The third, boosting-based method is applicable for a discrete approximation:

$$D_{c(i)}(P, Q) = \log 2 - L(F_i, P, Q); \tag{5}$$

$$F_i = F_{i-1} + \rho \cdot \arg\min_{f \in B} L(F_{i-1} + f, P, Q);$$

$$F_0 \equiv \frac{1}{2};$$

where: $\rho$ —learning rate, $B$ —base estimator, $c : \mathbb{Z}_+ \to [0, 1]$ —a strictly increasing function for mapping ensemble size onto $\alpha \in [0, 1]$.

Although Definition 2 is quite general, in this paper, we focus on families of pseudo-divergence produced in a manner similar to the examples above. All these examples introduce a classification algorithm parameterized by $\alpha$, then define pseudo-divergences $D_\alpha$ by substituting the optimal discriminator in Equation Eq. (2) with the discriminator trained in accordance with this classification algorithm with the parameter $\alpha$. Of course, one has to make sure that the resulting family of pseudo-divergences is ordered and complete w.r.t. Jensen–Shannon divergence. appendix provides formal definitions and proofs for the examples above.

With this class of pseudo-divergences in mind, we refer to $\alpha$ as capacity of the pseudo-divergence $D_\alpha \in \mathcal{D}$ relative to the family $\mathcal{D}$, or simply as capacity if the family $\mathcal{D}$ is clear from the context. In the examples above, capacity of pseudo-divergence is directly linked to the capacity of underlying discriminator models: to the size of the model in equation Eq. (3), to the strength of the regularization in equation Eq. (4) (which, similar to the previous case, effectively restricts the size of the set of feasible models) or to the size of the ensemble for a boosting-based family of divergences in equation Eq. (5).

Finally, we introduce a function that combines a family of pseudo-divergences into a single divergence.

**Definition 3** *If a family of pseudo-divergences* $\mathcal{D} = \{D_\alpha | \alpha \in [0,1]\}$ *is ordered and complete with respect to Jensen–Shannon divergence, then adaptive divergence* $\mathrm{AD}_\mathcal{D}$ *produced by* $\mathcal{D}$ *is defined as*:

$$\mathrm{AD}_\mathcal{D}(P,Q) = \inf\{D_\alpha(P,Q) | D_\alpha(P,Q) \geq (1-\alpha)\log 2\}. \tag{6}$$

We omit index in $\mathrm{AD}_\mathcal{D}$ when the family $\mathcal{D}$ is clear from the context or is not important.

A linear 'threshold' function $\tau(\alpha) = 1 - \alpha$ is used in the definition, however, it can be replaced by any strictly decreasing $\tau : [0,1] \to [0,1]$, such that $\tau(0) = 1$ and $\tau(1) = 0$:

$$\mathrm{AD}_\mathcal{D}(P,Q) = \inf\{D_\alpha(P,Q) | D_\alpha(P,Q) \geq \tau(\alpha)\log 2\}, \tag{7}$$

but, since one can redefine the family $\mathcal{D}$ as $\mathcal{D}' = \{D_{\tau(\alpha)} | \alpha \in [0,1]\}$, this effectively leads to the same definition. Nevertheless, it might be convenient in practice to use $\tau$ other than $\tau(\alpha) = 1 - \alpha$ as most model families have a natural ordering, e.g., regularization strength.

The coefficient $\log 2$ naturally arises as the maximal value of Jensen–Shannon divergence as well as an upper bound of any pseudo-divergence based on equation Eq. (3) if the function $f_0(x) = 1/2$ is included in the underlying classification model $M$. Since almost all popular models are capable of learning constant estimators, $\log 2$ is included in the definition. Nevertheless, to adopt Definition 3 for exotic models or divergences other than Jensen–Shannon (e.g., Wasserstein distance), this coefficient (and, possibly, the 'threshold' function) should be reconsidered.

Note, that due to property (**D1**), $\mathrm{D}_\alpha(P,Q)$ is a non-decreasing function of $\alpha$, while $(1-\alpha)\log 2$ is a strictly decreasing one. Hence, if family $\mathcal{D}$ is such that for any two distributions $P$ and $Q$ $\mathrm{D}_\alpha(P,Q)$ is continuous w.r.t. $\alpha$, equation Eq. (6) can be simplified:

$$\mathrm{AD}_\mathcal{D}(P,Q) = \mathrm{D}_{\alpha^*}(P,Q), \tag{8}$$

---

**Algorithm 1** General procedure for computing an adaptive divergence by grid search

---

**Require:** $\mathcal{D} = \{D_\alpha | \alpha \in [0,1]\}$ — ordered and complete w.r.t. Jensen-Shannon divergence family of pseudo-divergences; $\varepsilon$ — tolerance; $P, Q$ — input distributions

$\quad \alpha \leftarrow 0;$
$\quad$ **while** $D_\alpha(P,Q) < (1-\alpha)\log 2$ **do**
$\quad\quad \alpha \leftarrow \alpha + \varepsilon$
$\quad$ **end while**
$\quad$ **return** $D_\alpha(P,Q)$

---

where $\alpha^*$ is the root of the following equation:

$$D_\alpha(P,Q) = (1-\alpha)\log 2. \tag{9}$$

A general procedure for computing $\text{AD}_\mathcal{D}$ for this case is outlined in Algorithm 1.

Intuitively, an adaptive divergence $\text{AD}_\mathcal{D}$ switches between members of $\mathcal{D}$ depending on the 'difficulty' of separating $P$ and $Q$. For example, consider family $\mathcal{D}$ produced by equation Eq. (4) with a high-capacity neural network as model $M$ and $l_2$ regularization $R$ on its weights. For a pair of distant $P$ and $Q$, even a highly regularized network is capable of achieving low cross-entropy loss and, therefore, $\text{AD}_\mathcal{D}$ takes values of the pseudo-divergence based on such network. As distribution $Q$ moves close to $P$, $\text{AD}_\mathcal{D}$ lowers the regularization coefficient, effectively increasing the capacity of the underlying model.

The idea behind adaptive divergences can be viewed from a different angle. Given two distributions $P$ and $Q$, it scans the producing family of pseudo-divergences, starting from $\alpha = 0$ (the least powerful pseudo-divergence), and if some pseudo-divergence reports high enough value, it serves as a 'proof' of differences between $P$ and $Q$. If all pseudo-divergences from the family $\mathcal{D}$ report 0, then $P$ and $Q$ are equal almost everywhere as the family always includes JSD as a member. Formally, this intuition can be expressed with the following theorem.

**Theorem 1** *If* $\text{AD}_\mathcal{D}$ *is an adaptive divergence produced by an ordered and complete with respect to Jensen–Shannon divergence family of pseudo-divergences* $\mathcal{D}$, *then for any two distributions* $P$ *and* $Q$: $\text{JSD}(P,Q) = 0$ *if and only if* $\text{AD}(P,Q) = 0$.

A formal proof of Theorem 1 can be found in Appendix A2. Combined with the observation that $\text{AD}(P,Q) \geq 0$ regardless of $P$ and $Q$, the theorem states that AD is a divergence in the same sense as JSD. This, in turn, allows to use adaptive divergences as a replacement for Jensen–Shannon divergence in Adversarial Optimization.

As can be seen from the definition, adaptive divergences are designed to utilize low-capacity pseudo-divergences (with underlying low-capacity models) whenever it is possible: for a pair of distant $P$ and $Q$ one needs to train only a low-capacity model to estimate AD, using the most powerful model only to prove equality of distributions. As low-capacity models generally require fewer samples for training, AD allows an optimization algorithm to run for more iterations within the same time restrictions.

Properties of $\text{AD}_\mathcal{D}$ highly depend on the family $\mathcal{D}$, and choice of the latter might either negatively or positively impact convergence of a particular optimization algorithm.

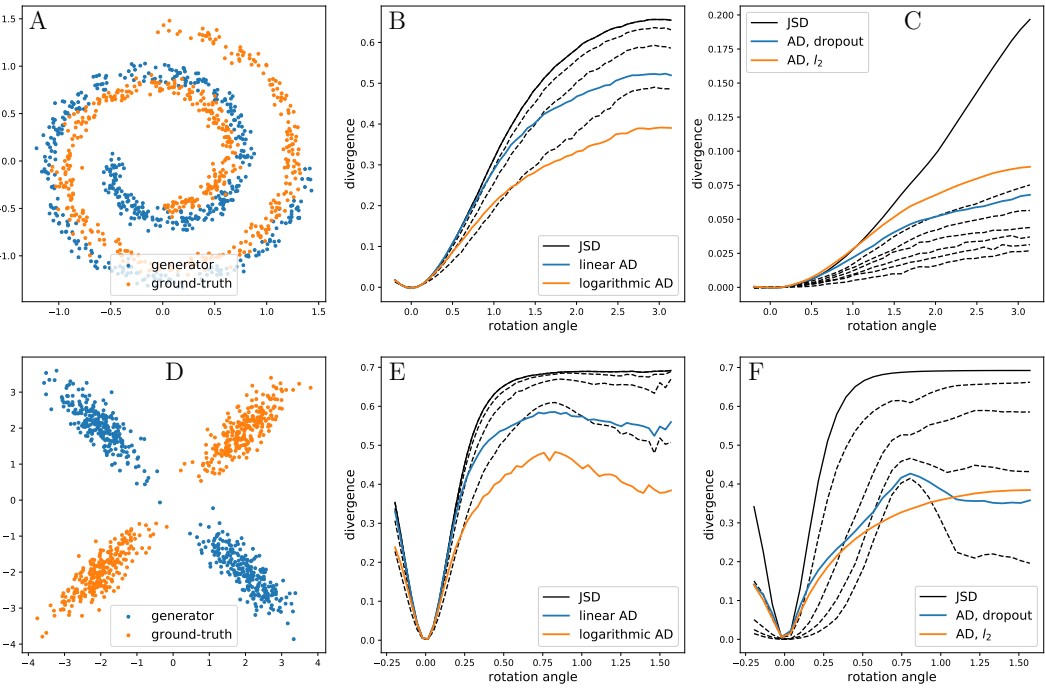

**Figure 1** **Synthetic examples.** (A) and (D): ground-truth distributions and example configurations of generators. Both generators are rotated versions of the corresponding ground-truth distributions. (B) and (E): JSD—Jensen–Shannon divergences estimated by Gradient Boosted Decision Trees with 500 trees of depth 3 (B), 100 trees of depth 3 (E); linear AD and logarithmic AD—adaptive divergences based on the same models as JSD with linear and logarithmic capacity functions, dashed lines represent some pseudo-divergences from the families producing adaptive divergences. (C) and (F): JSD —Jensen–Shannon divergences estimated by fully-connected Neural Networks with one hidden layer with 64 units (C) and 32 units (F); AD, dropout and AD, $l_2$—adaptive divergences based on the same architectures as the one for JSD, with dropout and $l_2$ regularizations; dashed lines represent some of the pseudo-divergences from the dropout-produced family. See 'Implementation' for the implementation details.

Figure 1 demonstrates both cases: here, we evaluate JSD and four variants of $\text{AD}_{\mathcal{D}}$ on two synthetic examples. In each example, the generator produces a rotated version of the ground-truth distribution and is parameterized by the angle of rotation (ground-truth distributions and examples of generator distributions are shown in Figs. 1A and 1D). In Figs. 1B and 1C AD shows behavior similar to that of JSD (both being monotonous and maintaining a significant slope in the respective ranges). In Fig. 1E, both variants of AD introduce an additional local minimum: as the rotation angle approaches $\pi/2$, marginal feature distributions become identical, which interferes with decision-tree-based algorithms (this is especially pronounced for AD with logarithmic capacity function as it prioritizes low-capacity models). This behavior is expected to impact convergence of gradient-based algorithms negatively.

In contrast, in Fig. 1F neural-network-based AD with $l_2$ regularization stays monotonous in the range $[0, \pi/2]$ and keeps a noticeable positive slope, in contrast to saturated JSD. The positive slope is expected to improve convergence of gradient-based algorithms and, possibly, some variants of Bayesian Optimization. In contrast, neural-network-based

AD with dropout regularization behaves in a manner similar to adaptive divergences in Fig. 1E. The most likely explanation is that $l_2$ regularization mostly changes magnitude of the predictions without significantly affecting the decision surface and, therefore, largely replicates behavior of JSD, while dropout effectively lowers the number of units in the network, which biases the decision surface towards a straight line (i.e., towards logistic regression).

## IMPLEMENTATION

A general algorithm for computing an adaptive divergence is presented in Algorithm 1. This algorithm might be an expensive procedure as the algorithm probes multiple pseudo-divergences, and for each of these probes, generally, a model needs to be trained from scratch. However, two of the most commonly used machine learning models, boosting-based methods (*Friedman, 2001*) and Neural Networks, allow for more efficient estimation algorithms due to the iterative nature of training procedures for such models.

### Gradient boosted decision trees

Gradient Boosted Decision Trees (*Friedman, 2001*) (GBDT) and, generally, boosting-based methods, being ensemble methods, intrinsically produce an ordered and complete with respect to Jensen–Shannon divergence family of pseudo-divergences in the manner similar to equation Eq. (5). This allows for an efficient AD estimation procedure shown by Algorithm 2. Here, the number of base estimators serves as capacity of pseudo-divergences, and mapping to $\alpha \in [0, 1]$ is defined through an increasing capacity function $c : \mathbb{Z}_+ \to [0, 1]$.[3]

In our experiments, for ensembles of maximal size $N$, we use the following capacity functions:

linear capacity: $\qquad c(i) = c_0 \dfrac{i}{N};$ (10)

logarithmic capacity: $\quad c(i) = c_0 \dfrac{\log(i+1)}{\log(N+1)}.$ (11)

Notice, however, that Equation Eq. (5) defines a discrete variant of AD, which most certainly will result in a discontinuous function.[4] This effect can be seen on Fig. 1E.

### Neural networks

There is a number of ways to regulate the capacity of a neural network. One of the simplest options is to vary the total number of units in the network. This, however, would almost certainly result in a discontinuous adaptive divergence, similarly to Gradient Boosted Decision Trees (Fig. 1E), which is not ideal even for black-box optimization procedures.

In this work, we instead use well-established dropout regularization *Srivastava et al. (2014)*. Effects of dropout are somewhat similar to varying number of units in a network, but at the same time dropout offers a continuous parametrization—it is clear that setting dropout probability $p$ to 0 results in an unregularized network, while $p = 1$ effectively restricts classifier to a constant output and intermediate values of $p$ produce models in between these extreme cases. To produce a family of pseudo-divergences we equip dropout

[3]Technically, this function should be extended on $[0, +\infty)$ to be in agreement with Definition 2.

[4]Note that introducing a continuous approximation of the ensemble by, for example, varying learning rate for the last base estimator in the current ensemble from 0 to $\rho$, eliminates discontinuity of AD.

---

**Algorithm 2** Boosted adaptive divergence

---

**Require:** $X_P, X_Q$ — samples from distributions $P$ and $Q$, $B$ — base estimator training algorithm, $N$ — maximal size of the ensemble, $c : \mathbb{Z}_+ \to [0, 1]$ — capacity function; $\rho$ — learning rate;

$F_0 \leftarrow 1/2$

$i \leftarrow 0$

$L_0 \leftarrow \log 2$

**for** $i = 1, \ldots, N$ **do**

    **if** $L_i > c(i) \log 2$ **then**

        $F_{i+1} \leftarrow F_i + \rho \cdot B(F_i, X_P, X_Q)$

        $L_{i+1} \leftarrow L(F_{i+1}, X_P, X_Q)$

        $i \leftarrow i + 1$

    **else**

        **return** $\log 2 - L_i$

    **end if**

**end for**

**return** $\log 2 - L_N$

---

regularization with a linear capacity function: $c(\alpha) = 1 - \alpha$, where $\alpha$ corresponds to dropout probability $p$.

Methods with explicit regularization terms can also be used to produce a family of pseudo-divergences. In this work, we examine $l_2$ regularization on network weights as one of the most widely used. In this case, a family of pseudo-divergences is defined by equation Eq. (4) with a logarithmic capacity function: $c(\alpha) = -\log(\alpha)$.

Regularization methods mentioned above were selected primarily due to their simplicity and popularity in the field. Our experiments indicate that these methods perform well. Nevertheless, further studies are required to determine best-performing regularization techniques.

In our experiments, we observe that unregularized networks require significantly more samples to be properly trained than regularized ones. To reduce discriminator variance, we suggest to use additional regularization $r$, strength of which is independent from the capacity parameter $\alpha$, e.g.:

$$D_\alpha(P, Q) = \log 2 - L(f(\theta^*, \cdot), P, Q); \tag{12}$$

$$\theta^* = \arg\min_{\theta \in \Theta} L(f(\theta, \cdot), P, Q) + c(1 - \alpha) \cdot R(\theta) + r(\theta).$$

In this work, following *Louppe, Hermans & Cranmer (2017)*, we use gradient regularization $r = R_1$ suggested by *Mescheder, Geiger & Nowozin (2018)*. Note, that such family of pseudo-divergences is no longer complete w.r.t Jensen–Shannon divergence, i.e., $D_1 \neq \text{JSD}$. Nevertheless, $D_1$ is still a proper divergence (*Mescheder, Geiger & Nowozin, 2018*) (which closely resembles JSD), and all results in this work hold with respect to such divergences including main theorems and claims, i.e., the family defined above still produces a (generalized) variant of adaptive divergence.

The proposed procedures for estimating AD is outlined in Algorithms 3 and 4. As chosen regularization methods result in families of pseudo-divergences continuous w.r.t $\alpha$, the proposed algorithm employs equation Eq. (8), i.e., it varies the strength of the regularization depending on the current values of the cross-entropy. The values of the loss function are estimated with an exponential moving average over losses on mini-batches during iterations of Stochastic Gradient Descent, with the idea that, for slowly changing loss estimations and small enough learning rate, network training should converge (*Liu, Simonyan & Yang, 2018*). We find that initializing exponential moving average with log2, which corresponds to the absent regularization, works best.

---

**Algorithm 3** Adaptive divergence estimation by a dropout-regularized neural network

---

**Require:** $X_P, X_Q$ — samples from distributions $P$ and $Q$;

$f_\theta : \mathcal{X} \times \mathbb{R} \to \mathbb{R}$ — neural network with parameters $\theta \in \Theta$, the second argument represents dropout probability and is zero if unspecified; $c$ — capacity function;

$\rho$ — exponential average coefficient;

$\beta$ — coefficient for $R_1$ regularization;

$\gamma$ — learning rate of SGD.

$L_{\mathrm{acc}} \leftarrow \log 2$
**while** not converged **do**
    $x_P \leftarrow \mathrm{sample}(X_P)$
    $x_Q \leftarrow \mathrm{sample}(X_Q)$
    $\zeta \leftarrow c\left(1 - \frac{L_{\mathrm{acc}}}{\log 2}\right)$
    $g_0 \leftarrow \nabla_\theta L(f_\theta(\cdot, \zeta), x_P, x_Q)$
    $g_1 \leftarrow \nabla_\theta \|\nabla_\theta f_\theta(x_P)\|^2$
    $L_{\mathrm{acc}} \leftarrow \rho \cdot L_{\mathrm{acc}} + (1 - \rho) \cdot L(f_\theta, x_P, x_Q)$
    $\theta \leftarrow \theta - \gamma \left(g_0 + \beta g_1\right)$
**end while**
**return** $\log 2 - L(f_\theta, X_P, X_Q)$

---

# EXPERIMENTS

Adaptive divergence was designed to require fewer samples than its conventional counterparts. However, for practical purposes, it is meaningless to consider this quantity outside the context of optimization. To illustrate this claim, consider the following divergence:

$$\mathrm{ID}(P, Q) = \begin{cases} 0, & \text{if } P = Q \text{ almost everywhere;} \\ 1, & \text{otherwise.} \end{cases}$$

Such divergence can be estimated in a manner similar to that of adaptive divergence: starting with a low-capacity model, train the model to distinguish between $P$ and $Q$, if the model reports any differences between distributions, return 1, otherwise increase the

capacity of the model and repeat, until a sufficiently high capacity is reached, in which case return 0. In terms of the number of samples, ID is expected to be more efficient than AD; at the same time, ID is a textbook example of intrinsically hard optimization problem, rendering it useless for Adversarial Optimization. Therefore, we judge the performance of adaptive divergence only within an optimization procedure.

Note that adaptive divergence is not expected to improve the optimization surface; nevertheless, as Fig. 1 demonstrates, the improvement is seemingly present in some instances; however, our experiments show that it does not play any significant role (see Appendix A3 for details). In the cases, when degradation of the optimization surface takes place, global optimization procedures, such as Bayesian Optimization, are still expected to benefit from the usage of AD by being able to perform more steps within the same budget on the number of generator calls.

We compare adaptive divergence against JSD on three tasks,[5] each task is presented by a parametrized generator, 'real-world' samples are drawn from the same generator with some nominal parameters. Optimization algorithms are expected to converge to these nominal parameters.

We evaluate the performance of adaptive divergences with two black-box optimization algorithms, namely Bayesian Optimization and Adversarial Variational Optimization. As computational resources spent by simulators are of our primary concern, we measure convergence of Adversarial Optimization with respect to the number of samples generated by the simulation, which is expected to be roughly proportional to the total time in case of computationally heavy simulations. We chose to neglect the time spent on training models as the proposed methods are intended for simulations that are significantly more computationally intensive than training of any model with a reasonable capacity, for example, running ATLAS simulation (*The ATLAS Collaboration, 2010*) for the same number of times as budgets in our experiments would require several years on a single-core CPU.

To measure the number of samples required to estimate a divergence, we search for the minimal number of samples such that the difference between train and validation losses is within $10^{-2}$ for Gradient Boosted Decision Trees and $5 \cdot 10^{-2}$ for Neural Networks.[6] As a significant number of samples is involved in loss estimation, for simplicity, we use point estimations of losses. For GBDT, we utilize a bisection root-finding routine to reduce time spent on retraining classifiers; however, for more computationally expensive simulators, it is advised to gradually increase the size of the training set until the criterion is met.

For each experiment, we report convergence plots—Euclidean distance from the current guess to the nominal parameters as a function of the number of examples generated by the simulator. As the performance of Bayesian Optimization is influenced by choice of the initial points (in our experiments, 5 points uniformly drawn from the search space), each experiment involving Bayesian Optimization is repeated 100 times, and aggregated results are reported. Similarly, experiments with Variational Optimization are repeated 20 times each.[6]

[5]Code of the experiments is available at https://github.com/HSE-LAMBDA/rapid-ao/

[6]This procedure requires generating an additional validation set of the size similar to that of the training set, which might be avoided by, e.g., using Bayesian inference, or cross-validation estimates.

---

**Algorithm 4** Adaptive divergence estimation by a regularized neural network

---

**Require:** $X_P$, $X_Q$ — samples from distributions $P$ and $Q$;

  $f_\theta : \mathcal{X} \to \mathbb{R}$ — neural network with parameters $\theta \in \Theta$;

  $R : \Theta \to \mathbb{R}$ — regularization function; $c$ — capacity function;

  $\rho$ — exponential average coefficient;

  $\beta$ — coefficient for $R_1$ regularization;

  $\gamma$ — learning rate of SGD.

  $L_{\mathrm{acc}} \leftarrow \log 2$
  **while** not converged **do**
      $x_P \leftarrow \mathrm{sample}(X_P)$
      $x_Q \leftarrow \mathrm{sample}(X_Q)$
      $\zeta \leftarrow c\left(1 - \frac{L_{\mathrm{acc}}}{\log 2}\right)$
      $g_0 \leftarrow \nabla_\theta \left[L(f_\theta, x_P, x_Q) + \zeta \cdot R(f_\theta)\right]$
      $g_1 \leftarrow \nabla_\theta \|\nabla_\theta f_\theta(x_P)\|^2$
      $L_{\mathrm{acc}} \leftarrow \rho \cdot L_{\mathrm{acc}} + (1 - \rho) \cdot L(f_\theta, x_P, x_Q)$
      $\theta \leftarrow \theta - \gamma \left(g_0 + \beta g_1\right)$
  **end while**
  **return** $\log 2 - L(f_\theta, X_P, X_Q)$

---

## XOR-like synthetic data

This task repeats one of the synthetic examples presented in Fig. 1D: ground truth distribution is an equal mixture of two Gaussian distributions, the generator produces a rotated version of the ground-truth distribution with the angle of rotation being the single parameter of the generator. The main goal of this example is to demonstrate that, despite significant changes in the shape of the divergence, global optimization algorithms, like Bayesian Optimization, can still benefit from the fast estimation procedures offered by adaptive divergences.

For this task, we use an adaptive divergence based on Gradient Boosted Decision Trees (100 trees with the maximal depth of 3) with linear and logarithmic capacity functions given by Eqs. (10) and (11) and $c_0 = 1/4$. Gaussian Process Bayesian Optimization with Matern kernel ($\nu = 3/2$ and scaling from $[10^{-3}, 10^3]$ automatically adjusted by Maximum Likelihood fit) is employed as optimizer.

Convergence of the considered divergences is shown in Fig. 2. As can be seen from the results, adaptive divergences tend to request fewer generator calls per estimation; and, given the same budget, both variants of adaptive divergence converge on parameters around an order of magnitude closer to the optimum than traditional JSD. Notice, that the initial rapid progress slows as optimizer approaches the optimum, and the slope of the curves becomes similar to that of JSD: this can be explained by AD approaching JSD as probed distributions become less distinguishable from the ground-truth one.

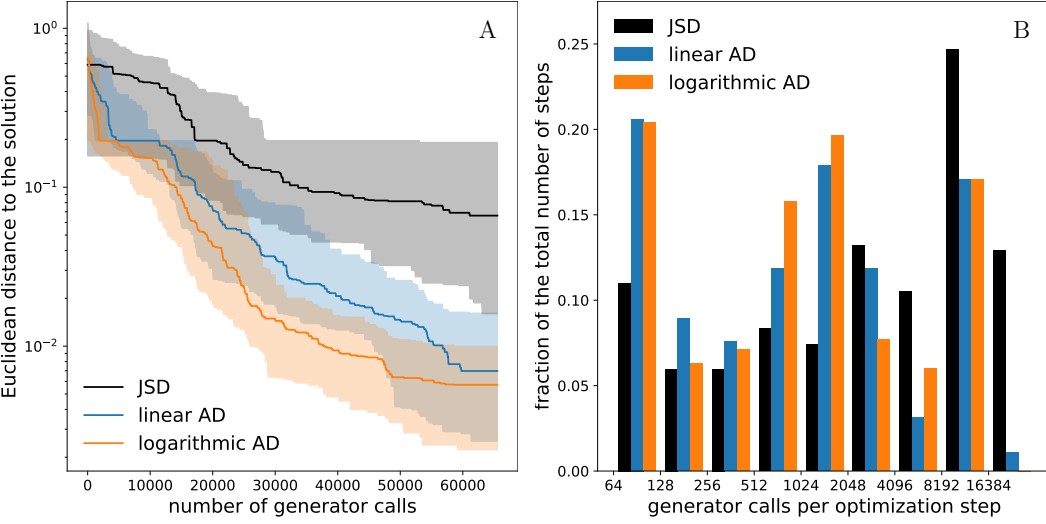

**Figure 2**  **XOR-like synthetic example, Gradient Boosted Decision Trees.** (A) Convergence of Bayesian Optimization on: Jensen–Shannon divergence (marked as JSD), adaptive divergences with a linear capacity function (marked as linear AD), and a logarithmic capacity function (logarithmic AD). Each experiment was repeated 100 times; curves are interpolated, median curves are shown as solid lines, bands indicate 25th and 75th percentiles. (B) Distribution of computational costs per single optimization step measured by the number of generator calls requested for divergence estimation; each optimization step requires exactly one divergence estimation; note logarithmic scaling of the $x$-axis.

[7]Note, that this task is rather a realistic toy example—in practical settings, Pythia event generator is followed by a much more computationally expensive detector simulation such as GEANT (*Allison et al., 2016*), the latter translates outcomes of an event generator, such as Pythia, into observable values. For comparison, full ATLAS simulation (event generator and detector simulation) mentioned above takes several minutes per sample, while Pythia alone typically require less than a second per event (milliseconds in our settings).

## Pythia hyper-parameter tuning

This task is introduced by *Ilten, Williams & Yang (2017)* and involves tuning hyper-parameters of the Pythia event generator, a high-energy particle collision simulation used at CERN. For this task, electron-positron collisions are simulated at a center-of-mass energy 91.2 GeV. As initial electron and positron collide and annihilate, new particles are created, some of which are unstable and might decay into more stable particles. A collision event is described by the properties of the final (stable) products. This process is intrinsically stochastic (due to the laws of physics) and covers a large space of possible outcomes, moreover, even with relatively large changes in generator's hyper-parameters, outcome distributions overlap significantly, which makes it an excellent example for adversarial optimization. The nominal parameters of the Pythia event generator are set to the values of the Monash tune (*Skands, Carrazza & Rojo, 2014*).

In work by *Ilten, Williams & Yang (2017)*, various physics-motivated statistics of events are used as observables,[7] with a total of more than 400 features. The same statistics were originally used to obtain the Monash tune. For the purposes of the experiment, we consider one hyper-parameter, namely alphaSValue, with the nominal value of 0.1365 and search range [0.06, 0.25].[7]

We repeat settings of the experiment[8] described by *Ilten, Williams & Yang (2017)*. We employ Gradient Boosting over Oblivious Decision Trees (CatBoost implementation by *Prokhorenkova et al., 2018*) with 100 trees of depth 3 and other parameters set to their default values. We use Gaussian Process Bayesian Optimization with Matern kernel ($\nu = 3/2$ and

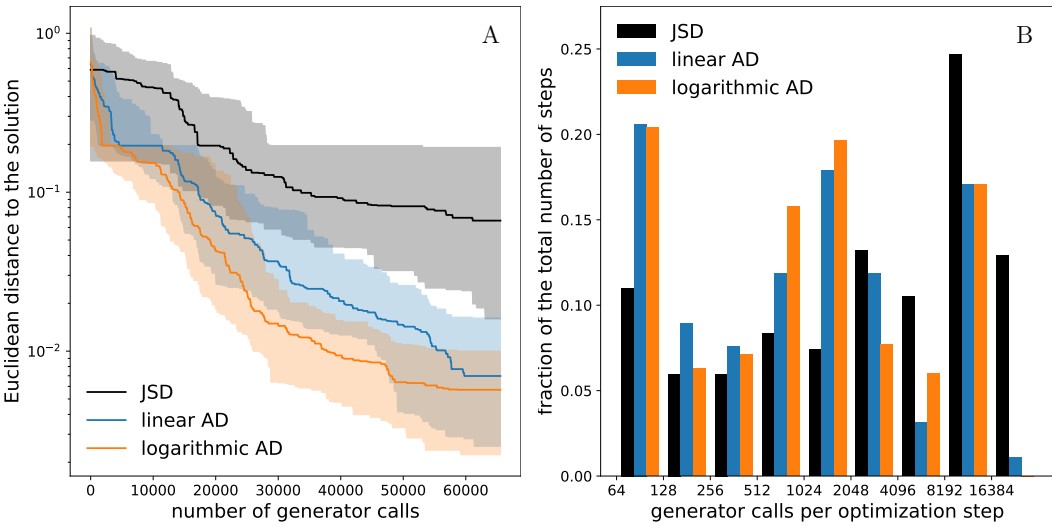

**Figure 3** **Pythia hyper-parameter tuning, CatBoost.** (A) Convergence of Bayesian Optimization on: Jensen–Shannon divergence (marked as JSD), adaptive divergences with a linear capacity function (marked as linear AD), and a logarithmic capacity function (logarithmic AD). Each experiment was repeated 100 times, curves are interpolated, median curves are shown as solid lines, bands indicate 25th and 75th percentiles. (B) Distribution of computational costs per single optimization step measured by the number of generator calls requested for divergence estimation; each optimization step requires exactly one divergence estimation; note logarithmic scaling of the $x$-axis.

[8]Methods proposed by *Ilten, Williams & Yang (2017)* compare a fixed set of statistics computed over multiple examples. As adversarial methods operate with individual examples, we use the same statistics computed for single events, i.e., original data can be recovered from ours by simply averaging across events.

scaling from $[10^{-3}, 10^3]$ automatically adjusted by Maximum Likelihood fit) as optimizer. Comparison of unmodified Jensen–Shannon divergence with adaptive divergences with linear and logarithmic capacity functions (defined by Eqs. (10) and (11) and $c_0 = 1/4$) presented on Fig. 3.[8]

Results, shown in Fig. 3, indicate that, given the same budget, Bayesian Optimization over adaptive divergences yields solutions about an order of magnitude closer to the nominal value than Jensen–Shannon divergence. This acceleration can be attributed to the proposed estimation procedures that require far fewer generator calls than JSD. Additionally, notice that the slope of the convergence curves for AD gradually approaches that of AD as the proposal distributions become closer to the ground-truth one.

## Pythia alignment

In order to test the performance of adaptive divergences with Adversarial Variational Optimization, we repeat the Pythia-alignment experiment suggested by *Louppe, Hermans & Cranmer (2017)*. The settings of this experiment are similar to the previous one. In this experiment, however, instead of collecting physics-motivated statistics, we consider a simplified detector simulation, represented by a $32 \times 32$ spherical grid with cells uniformly distributed in pseudorapidity $\nu \in [-5, 5]$ and azimuthal angle $\phi \in [-\pi, \pi]$ space. Each cell of the detector records the energy of particles passing through it. The detector has 3 parameters: $x, y, z$-offsets of the detector center relative to the collision point, where $z$-axis is placed along the beam axis, the nominal offsets are zero, and the initial

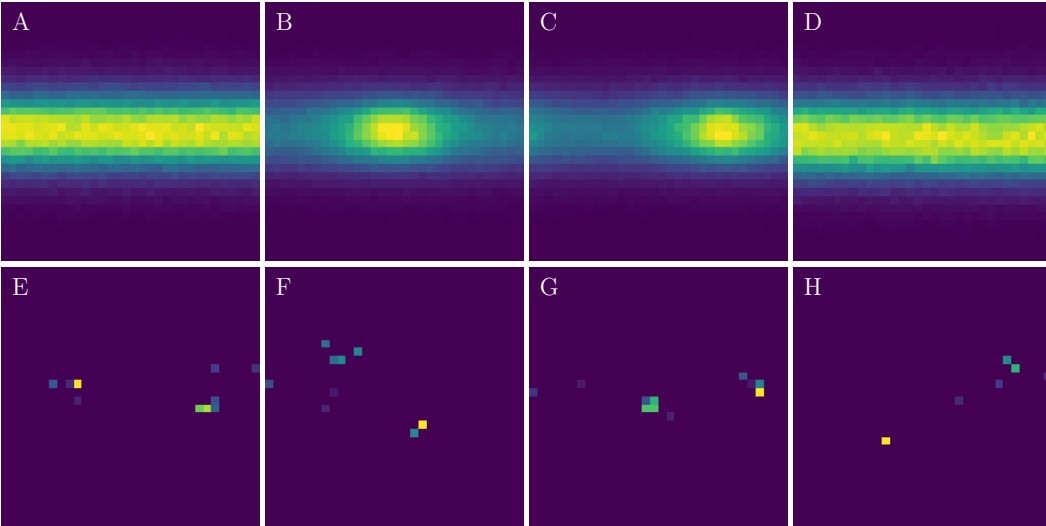

**Figure 4** **Illustration of the Pythia-alignment task.** (A) Aggregated events for zero offset (the nominal configuration), 0.25 offset along $x$-axis (B), $y$-axis (C), and $z$-axis (D). (E–H) Single-event examples from the corresponding configurations above—each activated pixel indicate a particle or multiple particles passing trough the corresponding region of the detector.

guess is $(0.75, 0.75, 0.75)$. Figure 4 shows averaged detector responses for the example configurations and samples from each of these configurations.

For this task, a 1-hidden-layer Neural Network with 32 hidden units and ReLU activation function is employed. $R_1$ regularization, proposed by *Mescheder, Geiger & Nowozin (2018)*, with the coefficient 10, is used for the proposed divergences and the baseline. Adam optimization algorithm (*Kingma & Ba, 2014*) with learning rate $10^{-2}$ is used to perform updates of the search distribution. We compare the performance of two variants of adaptive divergence (dropout and $l_2$ regularization) described in 'Implementation'.

Results are shown in Fig. 5. Adaptive divergences require considerably fewer samples for their estimation than the baseline divergence with only $R_1$ regularization, which, given the same budget, allows both variants of adaptive divergence to accelerate Adversarial Optimization significantly. Note that the acceleration is even more pronounced in comparison to JSD estimated by an unregularized network: in our experiments, to achieve the set level of agreement between train and test losses, the unregularized network often requires more samples than the entire budget.

## DISCUSSION

To the best knowledge of the authors, this work is the first one that explicitly addresses computational costs of Adversarial Optimization for expensive generators. Interestingly, several recent developments, like Progressive GAN (*Karras et al., 2017*) and ChainGAN (*Hossain et al., 2018*), use multiple discriminators of increasing capacity; however, this is done mainly to compensate for the growing capacity of the generators and, probably, not for reducing computational costs.

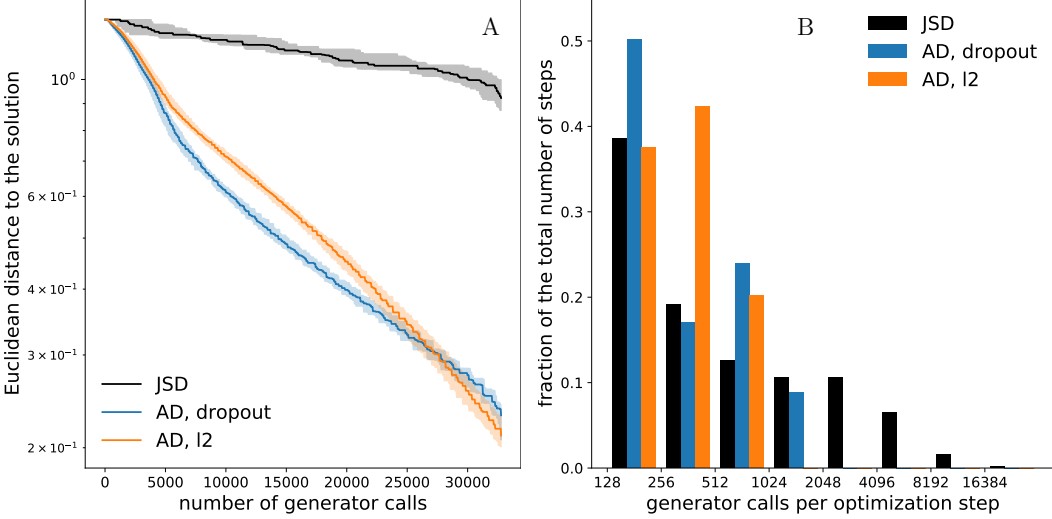

**Figure 5** **Pythia-alignment, neural networks.** (A) Convergence of Adversarial Variational Optimization on: adaptive divergence produced by $l_2$ regularization (AD, $l_2$), dropout regularization (AD, dropout), and the baseline divergence with constant $R_1$ regularization (marked as JSD). Each experiment was repeated 20 times, curves are interpolated, median curves are shown by solid lines, bands indicate 25th and 75th percentiles; steps-like patterns are interpolation artifacts. (B) Distribution of computational costs per single optimization step measured by the number of generator calls requested for divergence estimation; each optimization step requires exactly one divergence estimation; note logarithmic scaling of the $x$-axis.

Several recent papers propose improving stability of Adversarial Optimization by employing divergences other than Jensen–Shannon (*Gulrajani et al., 2017*; *Arjovsky, Chintala & Bottou, 2017*; *Bellemare et al., 2017*). Note that all results in this paper also hold for any divergence that can be formulated as an optimization problem, including Wasserstein (*Arjovsky, Chintala & Bottou, 2017*) and Cramer (*Bellemare et al., 2017*) distances. It can be demonstrated by adjusting Definition 2 and repeating the proof of Theorem 4 for a new divergence; presented algorithms also require only minor adjustments.

Multiple works introduce regularization (*Sønderby et al., 2016*; *Arjovsky, Chintala & Bottou, 2017*; *Roth et al. 2017*; *Kodali et al., 2017*; *Mescheder, Geiger & Nowozin, 2018*) for improving stability and convergence of Adversarial Optimization. Most of the standard regularization methods can be used to regulate model capacity in adaptive divergences. Also, one can use these regularization methods in addition to adaptive divergence as any discriminator-based regularization effectively produces a new type of divergence. Pythia-alignment experiment ('Pythia alignment') demonstrates it clearly, where we use $R_1$ regularization with constant coefficient in addition to varying-strength dropout and $l_2$ regularization.

As we discussed in 'Adaptive Divergence', properties of adaptive divergences highly depend on the underlying families of pseudo-divergences; the impact of various regularization schemes is a subject of future research.

## CONCLUSION

In this work, we introduce adaptive divergences, a family of divergences meant as an alternative to Jensen–Shannon divergence for Adversarial Optimization. Adaptive divergences generally require smaller sample sizes for estimation, which allows for a significant acceleration of Adversarial Optimization algorithms. These benefits were demonstrated on two fine-tuning problems involving Pythia event generator and two of the most popular black-box optimization algorithms: Bayesian Optimization and Variational Optimization. Experiments show that, given the same budget, adaptive divergences yield results up to an order of magnitude closer to the optimum than Jensen–Shannon divergence. Note, that while we consider physics-related simulations, adaptive divergences can be applied to any stochastic simulation.

Theoretical results presented in this work also hold for divergences other than Jensen–Shannon divergence.

## ACKNOWLEDGEMENTS

We wish to thank Mikhail Hushchyn, Denis Derkach, and Marceline Ivanovna for useful discussions and suggestions on the text.

## APPENDIX A1: FORMAL DEFINITIONS AND PROOFS

**Definition 4** *A model family* $\mathcal{M} = \{M_\alpha \subseteq \mathcal{F} | \alpha \in [0,1]\}$ *is complete and nested*, if:
**(N0)** $(x \mapsto 1/2) \in M_0$;
**(N1)**] $M_1 = \mathcal{F}$;
**(N2)** $\forall \alpha, \beta \in [0,1] : (\alpha < \beta) \Rightarrow (M_\alpha \subset M_\beta)$.

**Theorem 2** *If a model family* $\mathcal{M} = \{M_\alpha \subseteq \mathcal{F} | \alpha \in [0,1]\}$ *is complete and nested, then the family* $\mathcal{D} = \{D_\alpha : \Pi(\mathcal{X}) \times \Pi(\mathcal{X}) \to \mathbb{R} | \alpha \in [0,1]\}$, *where:*

$$D_\alpha(P, Q) = \log 2 - \inf_{f \in M_\alpha} L(f, P, Q), \tag{13}$$

*is a complete and ordered with respect to Jensen–Shannon divergence family of pseudo-divergences.*

**Proof** Let's introduce function $f_0(x) = 1/2$. Now we prove the theorem by proving that the family satisfies all properties from Definition 2.
**Property (D0)** Due to Properties (N0) and (N1), $f_0$ is a member of each set $M_\alpha$. This implies, that $D_\alpha(P, Q) \geq 0$ for all $\alpha \in [0,1]$. For $P = Q$, cross-entropy loss function $L(f, P, Q)$ achieves its minimum in $f = f_0$, therefore, $D_\alpha(P, Q) = 0$ if $P = Q$ for all $\alpha \in [0,1]$. Therefore, for each $\alpha \in [0,1]$ $D_\alpha$ is a pseudo-divergence.
**Property (D1)** From Properties (N2) follows, that for all $0 \leq \alpha < \beta \leq 1$:

$$D_\alpha(P, Q) = \log 2 - \inf_{f \in M_\alpha} L(f, P, Q) \geq \log 2 - \inf_{f \in M_\beta} L(f, P, Q) = D_\beta(P, Q).$$

**Property (D2)** This property is directly follows from Property (N1) and Equation Eq. (13).

**Definition 5** If $M$ is a parameterized model family $M = \{f(\theta,\cdot) : \mathcal{X} \to [0,1] | \theta \in \Theta\}$, then a function $R : \Theta \to \mathbb{R}$ is a proper regularizer for the family $M$ if:

**(R1)** $\forall \theta \in \Theta : R(\theta) \geq 0$;

**(R2)** $\exists \theta_0 \in \Theta : \left(f(\theta,\cdot) \equiv \frac{1}{2}\right) \wedge (R(\theta) = 0)$.

**Theorem 3** *If $M$ is a parameterized model family: $M = \{f(\theta,\cdot) | \theta \in \Theta\}$ and $M = \mathcal{F}$, $R : \Theta \to \mathbb{R}$ is a proper regularizer for $M$, and $c : [0,1] \to [0,+\infty)$ is a strictly increasing function such, that $c(0) = 0$, then the family $\mathcal{D} = \{D_\alpha : \Pi(\mathcal{X}) \times \Pi(\mathcal{X}) \to \mathbb{R} | \alpha \in [0,1]\}$:*

$$D_\alpha(P,Q) = \log 2 - \min_{\theta \in \Theta_\alpha(P,Q)} L(f(\theta,\cdot), P, Q);$$

$$\Theta_\alpha(P,Q) = \arg\min_{\theta \in \Theta} L_\alpha^R(\theta, P, Q);$$

$$L_\alpha^R(\theta, P, Q) = L(f(\theta,\cdot), P, Q) + c(1-\alpha)R(\theta);$$

*is a complete and ordered with respect to Jensen–Shannon divergence family of pseudo-divergences.*

**Proof** We prove the theorem by showing that the family $\mathcal{D}$ satisfies all properties from Definition 2.

**Property (D0)** Due to Properties (R2), there exists such $\theta_0$, that $f(\theta_0,\cdot) \equiv 1/2$ and $R(\theta_0) = 0$. Notice, that, for all $P$ and $Q$, $L_\alpha^R(\theta_0, P, Q) = \log 2$ and $L_\alpha^R(\theta, P, Q) \geq L(f(\theta,\cdot), P, Q)$, therefore, $D_\alpha(P,Q) \geq 0$ for all $P, Q \in \Pi(\mathcal{X})$ and for all $\alpha \in [0,1]$. For the case $P = Q$, $\theta_0$ also delivers minimum to $L(f(\theta_0,\cdot), P, Q) + c(1-\alpha)R(\theta_0)$, thus, $D_\alpha(P,Q) = 0$ if $P = Q$. This proves $D_\alpha$ to be a pseudo-divergence for all $\alpha \in [0,1]$.

**Property (D1)** Let's assume that $0 \leq \alpha < \beta \leq 1$, yet, for some $P$ and $Q$, $D_\alpha(P,Q) > D_\beta(P,Q)$. The latter implies, that:

$$\min_{\theta \in \Xi_\alpha} L(f(\theta,\cdot), P, Q) < \min_{\theta \in \Xi_\beta} L(f(\theta,\cdot), P, Q); \tag{14}$$

where: $\Xi_\alpha = \Theta_\alpha(P,Q)$ and $\Xi_\beta = \Theta_\beta(P,Q)$. Let us pick some model parameters:

$$\theta_\alpha \in \text{Arg}\min_{\theta \in \Xi_\alpha} L(f(\theta,\cdot), P, Q);$$

$$\theta_\beta \in \text{Arg}\min_{\theta \in \Xi_\beta} L(f(\theta,\cdot), P, Q).$$

Since $\theta_\beta \in \Xi_\beta$, then, by the definition of $\Theta_\beta(P,Q)$:

$$L_\beta^R(\theta_\beta, P, Q) \leq L_\beta^R(\theta_\alpha, P, Q). \tag{15}$$

From the latter and assumption (Eq. 14) follows, that $R(\theta_\beta) < R(\theta_\alpha)$. By the conditions of the theorem, $C = c(1-\alpha) - c(1-\beta) > 0$ and:

$$C \cdot R(\theta_\beta) < C \cdot R(\theta_\alpha). \tag{16}$$

Adding inequality Eq. (15) to inequality Eq. (16):

$$L_\alpha^R(\theta_\beta, P, Q) < L_\alpha^R(\theta_\alpha, P, Q),$$

which contradicts the definition of $\theta_\alpha$. This, in turn, implies that the assumption Eq. (14) contradicts conditions of the theorem.

**Property (D2)** Since $c(0) = 0$ and $M = \mathcal{F}$, $D_1 = \text{JSD}$ by the definition.

# APPENDIX A2. PROOF OF THEOREM 1

**Theorem 4** *If $AD_{\mathcal{D}}$ is an adaptive divergence produced by a complete and ordered with respect to Jensen–Shannon divergence family of pseudo-divergences $\mathcal{D}$, then for any two distributions $P$ and $Q$: $JSD(P,Q) = 0$ if and only if $AD(P,Q) = 0$.*

**Proof** For convenience, we repeat the definition of an adaptive divergence $AD_{\mathcal{D}}$ here:

$$AD_{\mathcal{D}}(P,Q) = \inf \left\{ D_\alpha(P,Q) | D_\alpha(P,Q) \geq (1-\alpha)\log 2 \right\}. \tag{17}$$

Firstly, we prove that from $JSD(P,Q) = 0$ follows $AD_{\mathcal{D}}(P,Q) = 0$. Due to Property (D2), $D_1(P,Q) = JSD(P,Q) = 0$, therefore, $\forall \alpha \in [0,1] : D_\alpha(P,Q) = 0$ due to Properties (D2) (pseudo-divergences form a non-decreasing sequence) and **(P1)** (non-negativity of pseudo-divergences), which, in turn, implies that $AD(P,Q) = \inf\{0\} = 0$.

Secondly, we prove that from $AD_{\mathcal{D}}(P,Q) = 0$ follows $JSD(P,Q) = 0$. Let's assume that, for some $P$ and $Q$, $AD(P,Q) = 0$, but $JSD(P,Q) = C > 0$. Let us define the set of active capacities $A_{\mathcal{D}}(P,Q)$ as follows:

$$A_{\mathcal{D}}(P,Q) = \left\{ \alpha | D_\alpha(P,Q) \geq (1-\alpha)\log 2 \right\}. \tag{18}$$

Note, that for every proper family $\mathcal{D}$ and for every pair of $P$ and $Q$: $\{1\} \subseteq A_{\mathcal{D}}(P,Q)$ and, if $\alpha \in A_{\mathcal{D}}(P,Q)$ then $[\alpha, 1] \subseteq A_{\mathcal{D}}(P,Q)$. The latter follows from Property (D1) (pseudo-divergences form a non-decreasing sequence) and the fact, that $(1-\alpha)\log 2$ is a strictly decreasing function. The previous statement implies that there are three possible forms of $A_{\mathcal{D}}(P,Q)$:

1. a single point: $A_{\mathcal{D}}(P,Q) = \{1\}$;
2. an interval: $A_{\mathcal{D}}(P,Q) = [\beta, 1]$;
3. a half-open interval: $A_{\mathcal{D}}(P,Q) = (\beta, 1]$;

for some $\beta \in [0,1)$. The first case would contradict our assumptions, since $AD_{\mathcal{D}}(P,Q) = \inf\{D_1(P,Q)\} = C > 0$. To address the last two cases, note, that $\forall \alpha \in A_{\mathcal{D}}(P,Q) : D_\alpha(P,Q) \geq (1-\beta)\log 2 > 0$ due to the definition of $A_{\mathcal{D}}(P,Q)$. However, this implies that $AD_{\mathcal{D}}(P,Q) = \inf\{D_\alpha(P,Q) | \alpha \in A_{\mathcal{D}}(P,Q)\} \geq (1-\beta)\log 2 > 0$, which contradicts our assumptions. From the statements above, we can conclude that if $AD_{\mathcal{D}}(P,Q) = 0$, then $JSD(P,Q) = 0$. Combined with the previouly proven $(JSD(P,Q) = 0) \Rightarrow (AD_{\mathcal{D}}(P,Q) = 0)$, this finishes the proof.

# APPENDIX A3. SOURCE OF THE ACCELERATION

Figures 2, 3 and 5 demonstrate that usage of adaptive divergence allows to accelerate Adversarial Optimization and lower requirements on the number of generator calls clearly play a major role. Nevertheless, this acceleration can be potentially attributed to the changes in the shape of the target function. Figure 6 shows convergence plots for the experiments described above; however, the $x$-axis corresponds to the optimization step rather than number of generator calls. These convergence plots demonstrate that changes in shape either do not affect convergence speed (Figs. 6A and 6B) or have a negative impact (Fig. 6C).

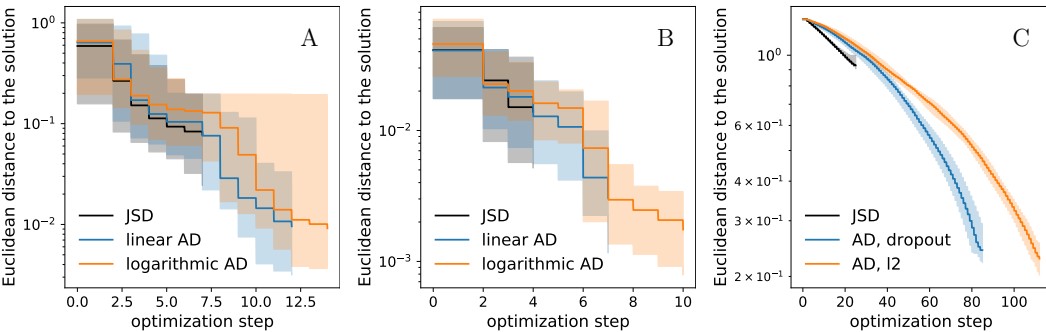

**Figure 6** **Convergence plots as functions of optimization step: (A) XOR-like synthetic dataset, (B) Pythia hyper-parameter tuning, (C) Pythia alignment.** Curves are interpolated, median curves are shown as solid lines, bars indicate 25th and 75th percentiles. For visual clarity curves are interpolated/extrapolated up to the median total number of steps for the corresponding method.

## Funding

The research leading to these results has received funding from Russian Science Foundation under grant agreement N 19-71-30020. The funders had no role in study design, data collection and analysis, decision to publish, or preparation of the manuscript.

## Grant Disclosures

The following grant information was disclosed by the authors:
Russian Science Foundation: 19-71-30020.

## Competing Interests

The authors declare there are no competing interests.

## Author Contributions

- Maxim Borisyak conceived and designed the experiments, performed the experiments, performed the computation work, prepared figures and/or tables, authored or reviewed drafts of the paper, and approved the final draft.
- Tatiana Gaintseva analyzed the data, performed the computation work, authored or reviewed drafts of the paper, and approved the final draft.
- Andrey Ustyuzhanin analyzed the data, prepared figures and/or tables, authored or reviewed drafts of the paper, and approved the final draft.

## Data Availability

The code for the experiments is available at https://github.com/HSE-LAMBDA/rapid-ao/.

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
