# Peer review of "Adaptive divergence for rapid adversarial optimization"

_PeerJ Computer Science, doi:10.7717/peerj-cs.274_

## Round 0.1 · original submission · Minor Revisions

The two reviewers are well qualified and appreciate the value of the paper. However, they have numerous suggestions for small improvements.

The requested changes will make the paper understandable to a significantly wider audience. They will be highly beneficial in terms of influencing other researchers and achieving the visibility that this research deserves.

When you submit the improved version of the paper, please include a separate statement that lists the changes requested by the two reviewers, and describes your response to each request. Please note that most of the changes requested are for additional explanations; the theme of the reviews is that the paper would benefit from being less terse.

·

Basic reporting

The paper is clear and well written. It does a good job of contextualizing the work w.r.t. to prior work. The paper is well structured, with results presented appropriately. The paper can be read and appreciated on its own. Theorems are presented clearly.

A section that could be improved is 5.2. For people unfamiliar with the Pythia program - which may be the majority of Machine Learning readers - this is not very understandable.

Experimental design

The paper is well structured, clearly defines a problem and the suggested approach. A sufficient number of experiments is run, though more would make for a stringer case. The methods are defined, though more details or looking at the source code release will be required for replicability.

I was able to download the code, however, I am not familiar with Pythia so could not test the code appropriately.

Validity of the findings

The findings have reasonable evidence as to their validity.

Since the code is provided and the results are simulation based, the findings should be robust and statistically sound.

Figure 5 seems to have no uncertainty intervals, as opposed to the other results. Was this run only once? Further experiments may be useful, though this step seems computationally expensive.

I am not sure if the results support the claimed hypothesis. While the paper mostly talks about reducing the number of required data points to be collected, the results support a different claim that AD improves the optimization surface so that better optima are found for the same number of data points. These claims are related but not the same.

Additional comments

The paper does not do a great job of discussing or explaining various choices made in the experiment. Overall, the paper is very interesting but it may be too terse in places and expanding on why choices were made or expanding on hypothesis as to why certain results were observed would make the paper much more valuable.

For example,
1. in D_{\alpha} where does the log 2 or the linear scaling with \alpha come from?
2. where exactly is the Bayesian Optimization coming in? I can kind of figure it out, but it isn't very clear from the definitions of e.g. Algorithm 3 and Algorithm 4.
3. The jump to dropout, layer norm and other optimization tricks (e.g. link 296) is stark, and many details are missing here. While I recognize that lots of tricks are required to get Adversarial models to train well, more details are discussion of the choices made here would be useful to the reader.
4. In the figures, it was not clear to me how 'distance to the solution' is defined. Is this the MSE between parameter configurations or JSD or something else?
5. Line 268 : with the only difference ... -> what is the significance of this difference?
6. Line 228 : as computational ... -> it would still be useful to comment on the other resources consumed and provide plots where possible.
7. Line 235 : As a significant ... -> This seems very important, and much of the validity may rest on this choice, but I do not understand it at all. Please expand here.
8. Line 296 : Note that .. -> This is an interesting conclusion and should be discussed in more detail.
9. Line 207 : dropout ... -> why this choice?
10. Line 216 : Note that a ... -> Is this true? Dropout ~ L2 regularization so I imagine it could still be a AD as defined in this paper.

Reviewer 2 ·

Basic reporting

The authors provide a thorough exposition into Adversarial methods for generative modeling and black-box optimization. In this work, Adversarial methods motivate methodologies where data gathering is expensive, necessitating a restriction to weaker discriminators that can be trained with fewer samples.

The authors identify the relationship between the Jensen-Shannon divergence and the objective function of the minimax objective function of Goodfellow’s original GAN paper. More specifically, given a perfect discriminator function, D, the generator, G, is tasked with minimizing the Jensen-Shannon divergence (JSD). (The reviewer would prefer this to be made more explicit in the paper.)

Experimental design

To demonstrate the monotonic scaling of pseudo-divergence with model capacity, the authors provide experimental results on a Gaussian-mixture model and the XOR dataset. (The referee would prefer there to be a brief discourse in observed cases where monotonicity is violated.)
The authors then use their approach to show improved sample-efficiency on hyper-parameter estimation in computationally expensive physics simulations. By replacing the JSD with the adaptive metric, the authors’ methods achieve much quicker convergence.

Validity of the findings

For a model with finite capacity, however, the cross-entropy loss can only approximate (Jensen-Shannon divergence – log(2)). In fact, the authors propose a family of pseudo-divergences that are parameterized by model capacity. Furthermore, the single-parameter divergence increases monotonically, limiting to JSD as this parameter approaches unity.
The novelty in this approach comes from the methodology in choosing a minimally expressive discriminator for a given pair of ground truth and generator distributions. The problem is posed such that a discriminator will always be found. Additionally, the search over discriminators starts by optimization of the lowest capacity model. As the estimation becomes more difficult, due to the increasing similarity of the two distributions, the network capacity of the newly trained discriminator increases.

Additional comments

Equation (1):
There should be a slightly expanded exposition about how optimal discriminator in the minimax game leads to the JSD. The second equality is a large jump that requires the definition:
D* = p_{data} / (p_{data} + p_{gen})
Section 5:
Are there any particular examples where the monotonicity does not generally hold? It seems this would occur when all discriminators yield pseudo-divergences near zero. It seems worth a few sentences, if there are concrete experiments yielding this behavior.

Figures 2 + 3:
What is the definition of “distance to the solution”? Is it the L2 norm of the difference between the ground truth and estimated vector of parameters? This should be made more clear.
Is the experimental time / FLOPS linearly proportional to “examples sampled” when comparing the different methodologies?

---

## Round 0.2 · Minor Revisions

Thank you for addressing the referees' comments. I would like to ask you also to expand the abstract to make it less terse. In the PDF attached, I have highlighted sentences from Sections 1, 2, and 7 that together would make a more comprehensible and detailed abstract. The article will be accepted after this improvement.

---

## Round 0.3 · accepted · Accept

Thank you for the improved abstract.